# The Contribution of Pharmacogenetic Drug Interactions to 90-Day Hospital Readmissions: Preliminary Results from a Real-World Healthcare System

**DOI:** 10.3390/jpm11121242

**Published:** 2021-11-23

**Authors:** Sean P. David, Lavisha Singh, Jaclyn Pruitt, Andrew Hensing, Peter Hulick, David O. Meltzer, Peter H. O’Donnell, Henry M. Dunnenberger

**Affiliations:** 1Department of Family Medicine, NorthShore University Health System, Evanston, IL 60201, USA; MDunnenberger@northshore.org; 2Department of Medicine, University of Chicago Pritzker School of Medicine, Chicago, IL 60637, USA; phulick@northshore.org (P.H.); dmeltzer@medicine.bsd.uchicago.edu (D.O.M.); podonnel@medicine.bsd.uchicago.edu (P.H.O.); 3Department of Statistics, NorthShore University Health System, Evanston, IL 60201, USA; lsingh@northshore.org; 4Department of Surgery, NorthShore University Health System, Evanston, IL 60201, USA; jpruitt@northshore.org; 5Outcomes Research Network, NorthShore University Health System, Evanston, IL 60201, USA; ahensing8@gmail.com; 6Center for Personalized Medicine, NorthShore University Health System, Evanston, IL 60201, USA

**Keywords:** pharmacogenetics, personalized, primary care

## Abstract

Clinical Pharmacogenetics Implementation Consortium (CPIC) guidelines exist for many medications commonly prescribed prior to hospital discharge, yet there are limited data regarding the contribution of gene-x-drug interactions to hospital readmissions. The present study evaluated the relationship between prescription of CPIC medications prescribed within 30 days of hospital admission and 90-day hospital readmission from 2010 to 2020 in a study population (N = 10,104) who underwent sequencing with a 14-gene pharmacogenetic panel. The presence of at least one pharmacogenetic indicator for a medication prescribed within 30 days of hospital admission was considered a gene-x-drug interaction. Multivariable logistic regression analyzed the association between one or more gene-x-drug interactions with 90-day readmission. There were 2211/2354 (93.9%) admitted patients who were prescribed at least one CPIC medication. Univariate analyses indicated that the presence of at least one identified gene-x-drug interaction increased the risk of 90-day readmission by more than 40% (OR = 1.42, 95% confidence interval (CI) 1.09–1.84) (*p* = 0.01). A multivariable model adjusting for age, race, sex, employment status, body mass index, and medical conditions slightly attenuated the effect (OR = 1.32, 95% CI 1.02–1.73) (*p* = 0.04). Our results suggest that the presence of one or more CPIC gene-x-drug interactions increases the risk of 90-day hospital readmission, even after adjustment for demographic and clinical risk factors.

## 1. Introduction

Despite a major national program to reduce hospital readmissions, 30-day hospital readmission rates for acute myocardial infarction, heart failure, and pneumonia are more than 17% [1], representing a major quality gap [2]. A recent systematic review found that 21% of hospital readmissions were due to adverse drug reactions [3]. Several large-scale pharmacogenetic implementation consortia have evaluated medical outcomes, but few have reported associations between a composite panel of gene-x-drug interactions and all prescribed medications that have evidence-based guidance from the Clinical Pharmacogenomics Implementation Consortium (CPIC) [4].

Studies from the Mayo Clinic Biobank have reported mixed results regarding associations between pharmacogenetic (PGx) phenotypes and hospital admissions [5,6], but these analyses did not evaluate gene-x-drug interactions. An Implementing GeNomics In pracTiCe (IGNITE) Network [7] multisite pragmatic investigation demonstrated that the risk of major adverse cardiovascular events for patients prescribed clopidogrel with a loss-of-function (LOF) *CYP2C19* allele was more than two-fold higher than patients without a LOF allele prescribed clopidogrel (adjusted hazard ratio (HR) 2.26, 95% CI 1.18–4.32) [8]. The Electronic Medical Records and Genomics (eMERGE) Network’s [9] multi-center pilot of electronic health record (EHR)-based pre-emptive pharmacogenetics (PGx) of the following gene–drug pairs (*CYP2C19*/clopidogrel, *CYP2C9*/*VKORC1*/warfarin, and *SLCO1B1*/simvastatin) [10] showed that patients prescribed warfarin within the CPIC dosing guideline range had a statistical trend toward faster time to first stable target international normalized ratio (INR) compared to a starting dose outside this range [11]. Although IGNITE and eMERGE have not yet reported hospitalization outcomes for most of their projects, there is evidence from smaller studies of pre-emptive PGx genotyping in elderly patients [12], patients with mental health disorders [13], and home health patients [14] that has shown reductions in hospitalizations [13,14], readmissions [14], and improvements in medication adherence [15].

The aim of our study was to evaluate the relationship between gene-x-drug interactions and the risk of 90-day hospital readmission. Given that there is no standard of care for pharmacogenetic medication tailoring prior to hospital discharge nationally, we anticipated that patients with gene-x-drug interactions would be more likely to have less optimal management and adherence, which over time would contribute to clinical manifestations requiring hospital readmission. Although there is not standard coding for every type of suboptimal drug response other than acute adverse drug reactions, we focused on the presence of actionable gene-x-drug interactions for these preliminary analyses. We hypothesized that patients with gene-x-drug interactions for medications prescribed up to and during hospital admission and discharge would be more likely to be readmitted within 90 days of hospital discharge.

## 2. Materials and Methods

### 2.1. Patient Population

The study population was primary care patients aged 18 or older who provided informed consent for genetic testing and who underwent Color™ (Burlingame, CA, USA) next-generation sequencing (NGS) for genetic screening of cancer and cardiovascular risk and a 14-gene pharmacogenetic panel between April 2019 and February 2020 [16]. Patients also provided written informed consent to have their clinical data utilized for operational purposes including quality review. Test results were reported in the electronic health record and reported to patients’ primary care physicians, and clinicians across the health system. Electronic medical records were searched for all hospital admissions from 2010 to 2020, medications with Clinical Pharmacogenetics Implementation Consortium guidelines (CPIC medications) prescribed within 30 days of hospital admission, and 90-day hospital readmissions and associated admission diagnoses, patient demographics, major chronic disease conditions, smoking, and COVID-19 status. Age, sociodemographic variables, and medical history represent the most recent available data extracted through to September 2021.

### 2.2. Data Extraction

Data on medication prescriptions were extracted for medications with accompanying 2021 CPIC guidelines at the time of writing. These medications included amitriptyline, atazanavir, atomoxetine, azathioprine, capecitabine, celecoxib, citalopram, clomipramine, clopidogrel, codeine, desipramine, doxepin, efavirenz, escitalopram, fluorouracil, flurbiprofen, fluvoxamine, fosphenytoin, ibuprofen, imipramine, lansoprazole, lornoxicam, meloxicam, mercaptopurine, nortriptyline, omeprazole, ondansetron, pantoprazole, paroxetine, peginterferon alfa-2a/2b, phenytoin, piroxicam, sertraline, simvastatin, tacrolimus, tamoxifen, tenoxicam, thioguanine, tramadol, trimipramine, tropisetron, voriconazole, and warfarin. Genetic indicators (genotype based phenotypes) for genes with evidence for variability in drug response were curated, annotated, and extracted for *CYP2B6*, *CYP2C19*, *CYP2C9*, *CYP2D6*, *CYP3A4*, *CYP3A5*, *CYP4F2*, *DPYD*, *F5*, *IFNL3*, *NUDT15*, *SLCO1B1, TPMT, and VKORC1*. Covariate data were extracted for sociodemographic variables (age, gender, ethnicity, employment, and marital status), and clinical variables (body mass index (BMI), smoking status, COVID-19 test results, history of cancer, chronic obstructive pulmonary disease (COPD), type 1 or 2 diabetes, myocardial infarction (MI), heart failure (HF), peripheral vascular disease (PVD), asthma, and stroke (cerebrovascular accident; (CVA)). For the purposes of this investigation, patients were considered to have a gene-x-drug interaction if they were prescribed at least one CPIC medication within 30 days of hospital admission for which they possessed a pharmacogenetic variant that portends altered drug pharmacokinetics, pharmacodynamics, and/or efficacy. ICD-10, if available, or ICD-9 diagnostic codes and conditions were binned into the following 21 categories for descriptive purposes: behavioral/psychiatric, cancer/neoplasm, cardiovascular, developmental/disability, endocrine/metabolic, hematological, gastrointestinal (GI), gynecological, infection/dermatological/abscess, infection/abscess/other, infection/respiratory, infection/GI, infection/genitourinary, inflammatory/rheumatological, obstetrical, orthopedic/musculoskeletal, otolaryngological, pain, pulmonary, renal, and vascular/non-cardiac.

### 2.3. Statistical Analysis

Clinical and demographic variables were compared between patients with 90-day readmissions and non-readmissions using a Wilcoxon rank sum test for continuous variables and a Chi-square or Fisher’s exact test for categorical variables. Patient descriptive statistics were reported as median and interquartile range (IQR) for continuous variables and frequency and percentage for categorical variables. Univariate and multivariable logistic regression analyses were performed to determine the association between drug interaction and 90-day readmission. Variables with statistically significant group differences and a *p*-value less than 0.1 in univariate analysis were included in multivariable logistic regression model as predictors of 90-day hospital readmission. Stepwise multivariable logistic regression was performed to determine a set of variables that best predicted the risk of 90-day hospital readmission in patients. Odds ratios (ORs) and corresponding 95% confidence intervals (CIs) were reported. *p*-values less than 0.05 were considered significant. All statistical analyses were performed using SAS version 9.4 (SAS Institute Inc., Cary, NC, USA) [17].

## 3. Results

A total of all 10,104 DNA10K patients were included in the analyses. Table 1 presents the main participant demographic and clinical characteristics of the study population. Of these individuals, there were 2354 participants (23.3%) with a history of at least one hospital admission from 1 January 2010 to 31 December 2020. Compared with patients who were not admitted, patients with a 10-year history of at least one hospital admission were more likely to be female (*p* < 0.0001), have higher BMI (*p* < 0.0001), more likely to be White/non-Hispanic (*p* = 0.0002), more likely to be married (*p* < 0.0001), less likely to be employed (*p* < 0.0001), and were more likely to have a history of a major medical condition including asthma, cancer, COPD, CVD, diabetes, HF, hypertension, MI, or PVD (*p* < 0.0001). There were no statistically significant differences in age, COVID-19 test results, or smoking status between patients with history of admission and no admissions from 2010 to 2020. Compared to patients with 10-year histories of at least one hospital admission who were not readmitted to hospital, patients with a history of 90-day hospital readmissions were older (median = 59, IQR: 44–67 years vs. median = 47, IQR: 37–62 years) (*p* < 0.0001), and readmitted patients were more likely to have a history of a major medical condition.

Of the 10,104 patients, 7885 (78%) were prescribed at least one CPIC medication at least once, and of the 2354 patients admitted between 2010 and 2020, 2333 (99.1%) were prescribed at least one CPIC medication. Of the patients admitted during this ten-year period, 2221 (93.9%) were prescribed at least one CPIC medication at least once, whereas only 9 (0.1%) of patients who were not admitted had been prescribed a CPIC medication over the last decade. Of the 281 patients readmitted to the hospital within 90 days, 276 (98.2%) were prescribed a CPIC medication within 30 days of their initial hospital admission compared with 1935/2073 (93.3%) of patients not readmitted to hospital within 90 days.

Appendix A lists CPIC medications prescribed within 30 days of initial hospital admission. The most frequently prescribed CPIC medications for patients readmitted within 90 days were ondansetron (n = 257, 93.1%), omeprazole (n = 125, 45.3%), ibuprofen (n = 101, 36.6%), tramadol (n = 95, 34.4%), warfarin (n = 52, 18.8%), celecoxib (n = 49, 17.8%), escitalopram (n = 27, 9.8%), simvastatin (n = 25, 9.1%), clopidogrel (n = 19, 6.9%), and codeine (n = 19, 6.9%). Sufficiently detailed data are not presently available to determine the relationship between medication prescribed and cause of hospital admission or readmission. There were 234 different diagnostic codes for 90-day hospital readmission. Appendix A reports admission diagnostic categories for patients with 90-day hospital readmissions. The most frequent admission diagnostic categories were endocrine/metabolic (n = 48, 17.4%), cancer/neoplasm (n = 32, 11.6%), infectious (n = 31, 13.9%), gastrointestinal (n = 30, 10.9%), obstetrical (n = 21, 7.6%), cardiovascular (n = 16, 5.8%), hematological (n = 16, 5.8%), orthopedic/musculoskeletal (n = 13, 4.7%), neurological (n = 10, 3.6%), or pulmonary (n = 10, 3.6%) conditions. Figure 1 is a flow diagram of patients included in the analyses based on admission and CPIC medication gene-x-drug interaction status.

Flow diagram of inclusion of patients in study population of whom all underwent genotyping with a next-generation sequencing Color™ (Burlingame, CA, USA) panel. Patients (n = 2211) who had a history of at least one hospital admission from 2010 to 2020 were included in the logistic regression analyses.

Table 2 presents results of multivariable logistic regression of the risk of 90-day hospital readmission (dependent variable) for patients prescribed a CPIC medication within 30 days of an initial hospital admission with at least one gene-x-drug interaction compared with prescription of at least one CPIC medication with no gene-x-drug interactions (reference). The unadjusted 90-day readmission rate for patients with a gene-x-drug interaction was 15.2% (107/705) compared with 11.2% (169/1506), (OR = 1.42, 95% CI 1.09–1.84) (*p* = 0.01). Age increasing per year (OR = 1.03, 95% CI 1.02–1.04) (*p* = <0.0001), male sex (OR = 1.76, 95% CI 1.34–2.33) (*p* = <0.0001), Black (OR = 1.98, 95% CI 1.20–3.25) (*p* = 0.005)) or Asian race (OR = 0.43, 95% CI 0.21–0.85) (*p* = 0.01), employment status (OR = 2.03, 95% CI 1.57–2.62) (*p* = 0.02), or the presence of three or more comorbidities (OR = 4.06, 95% CI 2.84–5.81) were also associated with higher rates of 90-day hospital readmission. Adjusting for sociodemographic variables and comorbidity, the odds of hospital admission for patients with at least one gene-x-drug interaction compared to no gene-x-drug interaction involving a CPIC medication prescribed within 30 days of initial hospital admission were slightly attenuated but statistically significant (OR = 1.32, 95% CI 1.02–1.73) (*p* = 0.04).

## 4. Discussion

To our knowledge, this is the first published study to report the net effect of a panel of pharmacogenetic variants on hospital readmission in a large, primary-care-based patient population. Most other studies have reported on results for special patient populations for specific gene-x-drug combinations. These data suggest that even though taking a CPIC medication may be a proxy of risk for hospital admission, the presence of one or more gene-x-drug interactions increases the risk of 90-day hospital readmissions above and beyond the risk contributed by the conditions under treatment by these medications.

This study has several limitations. Data are not available on all potential adverse drug reactions that result from suboptimal dosing or selection of medications because our data were limited to admission diagnoses, and some admission diagnoses (e.g., orthopedic conditions or bleeding) were not coded as inadequate pain control for a specific medication or bleeding resulting from dosing outside the recommended range for an anticoagulant based on genotype. Thus, we could not stratify the causes of hospital admission and readmission with specificity. Another limitation is that we did not have data curated for medication dose and its relationship to evidence-based dosing recommendations for a given genotype. Therefore, we were not able to determine adherence to CPIC prescribing guidelines. Moreover, there are numerous other factors that are known to affect drug response and adverse drug reactions that were not captured in the available data, ranging from hepatic and renal impairment to pharmacogenetic phenotypes. Furthermore, the complex relationships between genotypes that have inhibitory, induction, or phenoconversion effects on a range of medications (e.g., CYP2D6 metabolizer phenotypes), prescribed dose of relevant medications, and multiway gene-x-drug-x-drug interactions, the specific adverse events that could trigger hospital readmission, and outcomes were not possible to examine with the available data.

However, even with advanced physician decision support and guidance in our system [16,18,19], the presence of gene-x-drug interactions could theoretically affect drug response and risk of adverse events if not translated into tailored prescriptions at every point of the care continuum. Evidence is needed from real-world health systems to justify the system-level investment in high-level pharmacogenetic decision support and patient monitoring. These results may contribute to this body of evidence, but additional investigations will need to address limitations in our present data using additional tools of data integration and innovation such as machine learning, natural language processing, and artificial intelligence [20] to capture and conduct more sensitive phenotyping of potential adverse drug reactions and discordant dosing for specific genotypes over time. However, results from smaller studies noted earlier of special patient populations have suggested that pre-emptive pharmacogenetic genotyping can affect more favorable outcomes for a number of patient populations including the elderly [12], patients with mental health disorders [13], and home health patients [14] that have shown reductions in hospitalizations [13,14], readmissions [14], and improvements in medication adherence [15].

## 5. Conclusions

This study showed an association between gene-x-drug interactions and the risk of 90-day hospital admissions for a general patient population amongst patients prescribed one or more medications with evidence for genetic variability in drug response. Replication and deeper phenotyping are needed to provide more clinical context for these observations. However, the effect size observed for this difference, if taken to scale, indicates that gene-x-drug interactions may contribute substantially to the risk of preventable hospital admissions. Future prospective studies are needed to evaluate the potential of pre-emptive genotyping and pharmacogenetic tailoring of patients at risk for hospital admission and readmission as a potentially effective intervention to prevent the morbidity and suboptimal treatment of clinical conditions following hospital discharge that can lead to hospital readmission.

## Figures and Tables

**Figure 1 jpm-11-01242-f001:**
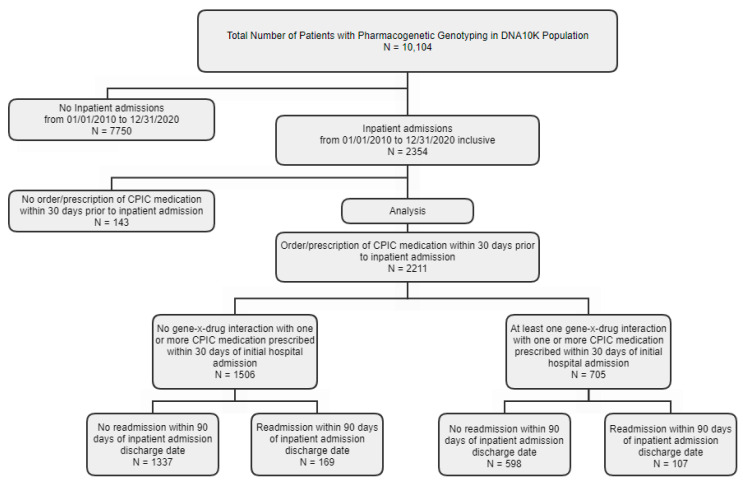
Flow diagram of patient selection for analyses.

**Table 1 jpm-11-01242-t001:** Demographic and clinical characteristics of DNA10K patient population.

	Inpatient Admissions 2010–2020 ^a^						
	Overall 10,104 (100)	No 7750 (76.7)	Yes 2354 (23.3)		90-Day Hospital Readmission ^b^		
					N = 2354		
					No 2073 (88.1)	Yes 281 (11.9)	
**Age, median (IQR)**	50 (39–60)	50 (40–60)	48 (38–63)		47 (37–62)	59 (44–67)	*
**BMI, median (IQR)**	26.7 (23.6–31.1)	26.6 (23.5–30.8)	27.5 (23.8–32.1)	*	27.3 (32.7–31.9)	28.4 (24.5–33.3)	*
**Sex**				*			*
Female	6684 (66.2)	4870 (62.8)	1,814 (77.1)		1622 (78.2)	192 (68.3)	
Male	3420 (33.8)	2880 (37.2)	540 (22.9)		451 (21.8)	89 (31.7)	
**Race**				*			*
White	6845 (67.7)	5117 (66)	1728 (73.4)		1510 (72.8)	218 (77.6)	
Black or African American	304 (3)	201 (2.6)	103 (4.4)		80 (3.9)	23 (8.2)	
Asian	746 (7.4)	586 (7.6)	160 (6.8)		151 (7.3)	9 (3.2)	
American Indian or Alaska Native	23 (0.2)	19 (0.2)	4 (0.2)		4 (0.2)	0 (0)	
Pacific Islander/Hawaiian Native	3 (0)	3 (0)	0 (0)		0 (0)	0 (0)	
Other	2,084 (20.6)	1737 (22.4)	347 (14.7)		318 (15.3)	29 (10.3)	
Declined/Unknown	99 (1)	87 (1.1)	12 (0.5)		10 (0.5)	2 (0.7)	
**Ethnicity**				*			
Hispanic/Latino	497 (4.9)	374 (4.8)	123 (5.2)		112 (5.4)	11 (3.9)	
Non-Hispanic	9480 (93.8)	7259 (93.7)	2221 (94.4)		1952 (94.2)	269 (95.7)	
Declined/Unknown	127 (1.3)	117 (1.5)	10 (0.4)		9 (0.4)	1 (0.4)	
**Current Patient Status**				*			*
Alive	10,083 (99.8)	7743 (99.9)	2340 (99.4)		2067 (99.7)	273 (97.2)	
Deceased	21 (0.2)	7 (0.1)	14 (0.6)		6 (0.3)	8 (2.8)	
**Marital Status**				*			*
Unmarried	2823 (27.9)	2232 (28.8)	591 (25.1)		492 (23.7)	99 (35.2)	
Married	7198 (71.2)	5439 (70.2)	1759 (74.7)		1577 (76.1)	182 (64.8)	
Unknown	83 (0.8)	79 (1)	4 (0.2)		4 (0.2)	0 (0)	
**Employee Status**				*			*
Unemployed	2899 (28.7)	2059 (26.6)	840 (35.7)		697 (33.6)	143 (50.9)	
Employed	6897 (68.3)	5443 (70.2)	1454 (61.8)		1323 (63.8)	131 (46.6)	
Unknown	308 (3)	248 (3.2)	60 (2.5)		53 (2.6)	7 (2.5)	
**Smoking Status**							
No	9751 (96.5)	7467 (96.3)	2284 (97)		2015 (97.2)	269 (95.7)	
Yes	353 (3.5)	283 (3.7)	70 (3)		58 (2.8)	12 (4.3)	
**COVID Status**							
Yes	52 (0.5)	39 (0.5)	13 (0.6)		12 (0.6)	1 (0.4)	
**Comorbidities**							
Cancer	101 (1)	44 (0.6)	57 (2.4)	*	41 (2)	16 (5.7)	*
COPD	98 (1)	44 (0.6)	54 (2.3)	*	40 (1.9)	14 (5)	*
Diabetes	671 (6.6)	442 (5.7)	229 (9.7)	*	182 (8.8)	47 (16.7)	*
History of Diabetes	740 (7.3)	479 (6.2)	261 (11.1)	*	207 (10)	54 (19.2)	*
Myocardial Infarction	50 (0.5)	17 (0.2)	33 (1.4)	*	26 (1.3)	7 (2.5)	
Heart Failure	93 (0.9)	37 (0.5)	56 (2.4)	*	36 (1.7)	20 (7.1)	*
Hypertension	2074 (20.5)	1463 (18.9)	611 (26)	*	491 (23.7)	120 (42.7)	*
PVD	83 (0.8)	35 (0.5)	48 (2)	*	35 (1.7)	13 (4.6)	*
Asthma	940 (9.3)	655 (8.5)	285 (12.1)	*	237 (11.4)	48 (17.1)	*
CVA	239 (2.4)	115 (1.5)	124 (5.3)	*	91 (4.4)	33 (11.7)	*
**CPIC medication order/admin within 30 days prior to admission date in electronic medical record between 12/01/2009 and 12/31/2020 inclusive (N = 2211)**					**1935/2211 (87.5)**	**276/2211 (12.5)**	
**Drug interactions with orders in med Orders/Admin within 30 days prior to inpatient admission**							*
Absent					1337 (69.1)	169 (61.2)	
Present					598 (30.9)	107 (38.8)	

^a^ Inpatient hospital admissions between 1 January 2010 to 31 December 2020 inclusive. ^b^ Readmissions within 90 days of inpatient admission discharge date between 1 January 2010 and 31 March 2021 inclusive. * *p*-values less than 0.05. Cerebrovascular accident (CVA). Clinical Pharmacogenetics Implementation Consortium (CPIC) (https://cpicpgx.org/, accessed on 22 November 2021). Chronic obstructive pulmonary disease (COPD). Interquartile range (IQR). Peripheral vascular disease (PVD).

**Table 2 jpm-11-01242-t002:** Unadjusted and adjusted logistic regression analysis of gene-x-drug interaction and association with 90-day readmission among patients prescribed a CPIC medication within 30 days of hospital admission (N = 2211).

	Univariate Regression		Multivariable Regression Using Stepwise Selection Method	
	uOR (95% CI)	*p*-Value	aOR (95% CI)	*p*-Value
Drug Interactions with Orders in Medication Orders/Administration within 30 Days Prior to Inpatient Admission Inclusive				
No	Reference	0.01	Reference	0.04
Yes	1.42 (1.09–1.84)		1.32 (1.01–1.73)	
**Age (continuous, range 17–90)**	1.03 (1.02–1.04)	<0.0001	1.02 (1.01–1.03)	0.0003
**Sex**				
Female	Reference		-	-
Male	1.76 (1.34–2.33)	<0.0001	-	-
**Race**				
Caucasian	Reference		Reference	
Black or African American	1.98 (1.20–3.25)	0.005	1.84 (1.09–3.11)	0.09
Asian	0.43 (0.21–0.85)	0.01	0.57 (0.28–1.14)	0.03
Other *	0.60 (0.40–0.91)	0.06	0.75 (0.49–1.14)	0.07
Declined/Unknown	1.46 (0.31–6.79)	0.49	2.73 (0.58–12.99)	0.18
**Ethnicity**				
Non-Hispanic	Reference		-	-
Hispanic/Latino	0.72 (0.38–1.35)	0.67	-	-
Declined/Unknown	0.86 (0.11–6.92)	0.99	-	-
**Most recent BMI ≤ 30**				
Yes	Reference		-	-
No	1.30 (1.00–1.68)	0.05	-	-
**Marital Status**				
Unmarried	Reference		-	-
Married	0.57 (0.43–0.74)	0.98	-	-
**Employee Status** **				
Employed	Reference		-	-
Unemployed	2.03 (1.57–2.62)	0.02	-	-
Unknown	1.39 (0.61–3.12)	0.95	-	-
**Smoking Status**				
No	Reference		-	-
Yes	1.68 (0.88–3.19)	0.11	-	-
**Number of comorbidities** ***				
0	Reference		Reference	
1	2.02 (1.48–2.77)	0.64	1.66 (1.19–2.30)	0.91
2	2.53 (1.65–3.88)	0.27	1.78 (1.13–2.82)	0.71
3 or more	4.06 (2.84–5.81)	<0.0001	2.69 (1.79–4.03)	0.001

uOR: unadjusted odds ratio, aOR: adjusted odds ratio, CI: confidence interval. * Other race includes American Indian or Alaska Native, Pacific Islander/Hawaiian Native, and other races. ** Employed includes full time, part time, and self-employed; unemployed includes not employed, retired, and students. *** Comorbidities include cancer, COPD, diabetes, diabetes PL/HX, myocardial infarction, heart failure, hypertension, peripheral vascular disease (PVD), asthma, and CVA. *p*-values less than 0.05 were considered statistically significant. Admission dates from electronic health record 1 December 2009 and 31 December 2020 inclusive.

## Data Availability

Data relevant to the study have been included in the article. Further data are available from the authors upon request.

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
