# Peer review of "The Contribution of Pharmacogenetic Drug Interactions to 90-Day Hospital Readmissions: Preliminary Results from a Real-World Healthcare System"

_jpm, 2021, doi:10.3390/jpm11121242_

Round 1

Reviewer 1 Report

Dear Authors,

Pharmacogenetics is a rapidly growing field studying of how genetic differences influence the variability of individual patient responses to drugs, aims to distinguish responders from non-responders and predict those in whom toxicity.

ADRs can result from inappropriate drug prescription, toxic effects of drug chemicals, impaired absorption, distribution, metabolism, and elimination of drugs related to age and sex, drug-drug interactions in combination therapy, or when a patient is treated with different medications for comorbid disorders This is especially important in chronic diseases that require long-term treatment and the treatment of the elderly, who, take several types of drugs daily  Today PGx is not a routine in clinical practice which may explain the lack of statistically significant data about the underlying reasons for ADRs-caused mortality. However, it is well known that the majority of ADRs are dose-dependent while the rest are related to allergy or idiosyncratic Usage of anticoagulants, opioids, or immunosuppressants is the most frequent reason for the lethal outcome.

In addition to genetics, as metioned above many other factors influence drug metabolism, such as sex, age, nutritional condition, hormonal and diurnal influences, concomitant drug use, or (underlying) diseases. Accordingly, Phenoconversion is the mismatch between the individual’s genotype-based prediction of drug metabolism and the true capacity to metabolize drugs due to nongenetic factors. Studies demonstrated that phenoconversion into a lower metabolizer phenotype is because of concomitant use of CYP450-inhibiting drugs, increasing age, cancer, and inflammation. Unfortunately , in current clinical practice, genetic factors are ignored when interpreting drug–drug interactions (DDIs). For example,  intermediate metabolizer (IM) as indicative of a give genotype is more susceptible to phenoconversion  than normal metabolizers (NM)  

In this article though Sean and his colleagues did not speak the importance of and phenoconversion effect during drug x gen interaction (p, the demonstrate the important effect of drug x gen interaction in hospital admission by establishing a detailed classification of patients

To this end, this manuscript highlights the potential opportunities and applications of PGx in clinical practice for personalized drug dosing for an effective pharmacotherapy. 

The manuscript is very well written, and the introduction provides a good, generalized background of the topic that quickly gives the reader an appreciation of the applications. Moreover, the topic and the data are very interesting. Certainly, this study will contribute much to the literature.

Thank you

Best Regards

Author Response

We appreciate and agree with the reviewer’s thoughtful comments and salient points. With respect to the comment that we did not speak to the importance of the "phenoconversion effect during drug-x-gene interaction…", we have now added the following sentences to the Discussion on page 9.

“Moreover, there are numerous other factors that are known to affect drug response and adverse drug reactions that were not captured in the available data ranging from hepatic and renal impairment to pharmacogenetic phenotypes. Furthermore, the complex relationships between genotypes that have inhibitory, induction, or phenoconversion effects on a range of medications (e.g., CYP2D6 metabolizer phenotypes), prescribed dose of relevant medications, and multiway gene-x-drug-x-drug interactions, the specific adverse events that could trigger hospital readmission, and outcomes were not possible to examine with the available data.”

We acknowledge that the effects of phenoconversion of drugs that is genetically influenced is an important factor in risk of adverse drug events and one that we were not able to capture with the depth of the available data.

Reviewer 2 Report

This is a very interesting manuscript describing the association between one or more gene-x-drug interactions with 90-day hospital readmission. The methodology used is appropriate, the results well presented and the conclusions supported by the data reported.

The main concern of this manuscript is the unclear description about how the Adjusted Logistic Regression Analysis was calculated. Thus, it will be necessary to clarify this. In addition, the authors should clarify how is possible to have an adjusted odds ratio p-value that is lower than the unadjusted odds ratio p-value, such as the unknowns in Table 2. 

Furthermore, a stepwise forward conditional and/or backward conditional multivariate logistic regression model should be performed to avoid confounding effects.

Author Response

We appreciate the reviewer’s comments about the manuscript. The reviewer raised critiques on two issues:

1) Unstable p-values: “The authors should clarify how it is possible to have an adjusted odds ratio p-value that is lower than the unadjusted odds ratio p-value, such as the unknowns in Table 2."

A probable explanation for this observation is that the unknown race category only has 12 participants (2 readmissions vs 10 no readmission). The 95% confidence interval is somewhat wider (0.58-12.99) for the adjusted analyses of declined/unknown race category than it is for the unadjusted analyses (0.31-6.79). We believe that this phenomenon for this variable is the result of the small cell size for the adjusted analyses results in unstable point estimates, a wider confidence interval and therefore a lower, but not statistically significant p-value. P-values for other variables did not follow this pattern.

2) Description of how the adjusted logistic regression analysis was performed: We acknowledge that more detail is needed to describe the multivariable logistic regression analyses. We reported results in the original manuscript that were derived from an initial univariate analysis that identified variables with statistically significant group differences and p-value less than 0.1, followed by a multivariable logistic regression model that included robust variables from the univariate analysis as predictors of 90-day hospital readmission. In response to the reviewer’s suggestion, we conducted stepwise multivariable logistic regression to determine a set of variables that best predict the risk of 90-days hospital readmission in patients – noted in revised Methods section on p. 3. The results in Table 2 have been updated with the stepwise multivariable analysis, which did not alter the effect size or p-values of the unadjusted and adjusted analyses.

We hope that these explanations and revisions have suitably addressed these important points.